# The Inflammatory Profile Orchestrated by Inducible Nitric Oxide Synthase in Systemic Lupus Erythematosus

**DOI:** 10.3390/jpm13060934

**Published:** 2023-05-31

**Authors:** Corina Daniela Ene, Ilinca Nicolae

**Affiliations:** 1Internal Medicine and Nephrology Department, Carol Davila University of Medicine and Pharmacy, 050474 Bucharest, Romania; 2Nephrology Department, Carol Davila Clinical Hospital of Nephrology, 010731 Bucharest, Romania; 3Dermatology Department, Victor Babes Clinical Hospital of Tropical and Infectious Diseases, 030303 Bucharest, Romania; drnicolaei@yahoo.ro

**Keywords:** systemic lupus erythematosus, inducible nitric oxide synthase, inflammatory phenotype, angiogenesis

## Abstract

(1) Background: The pathogenesis of systemic lupus erythematosus (SLE) involves complicated and multifactorial interactions. Inducible nitric oxide synthase overactivation (iNOS or NOS2) could be involved in SLE pathogenesis and progression. This study explored the relationship between NOS2-associated inflammation profiles and SLE phenotypes. (2) Methods: We developed a prospective, case control study that included a group of 86 SLE subjects, a group of 73 subjects with lupus nephritis, and a control group of 60 people. Laboratory determinations included serum C reactive protein (CRP–mg/L), enzymatic activity of NOS2 (U/L), serum levels of inducible factors of hypoxia 1 and 2 (HIF1a–ng/mL, HIF2a–ng/mL), vascular endothelial growth factor VEGF (pg/mL), matrix metalloproteinases 2 and 9 (MMP-2, MMP-9–ng/mL), thrombospondin 1 (TSP-1–ng/mL), and soluble receptor of VEGF (sVEGFR–ng/mL). (3) Results: CRP, NOS2, HIF-1a, HIF-2a, VEGF, MMP-2, and MMP-9 were significantly increased, while TSP-1 and sVEGFR were decreased in the SLE and lupus nephritis groups compared with the control group. The variations in these biomarkers were strongly associated with the decrease in eGFR and increase in albuminuria. (4) Conclusions: The inflammatory phenotype of SLE patients, with or without LN, is defined by NOS2 and hypoxia over-expression, angiogenesis stimulation, and inactivation of factors that induce resolution of inflammation in relation with eGFR decline.

## 1. Introduction

The pathogenesis of systemic lupus erythematosus (SLE) involves complicated and multifactorial interactions. Inflammation, oxidative imbalance, immune activation, autoantibody production, overstimulation of type 1 interferon (IFN), generation of inflammatory mediators, and tissue damage are fundamental contributors to SLE pathogenesis [1,2,3]. A series of experimental data have suggested that reactions mediated by inducible synthase of nitric oxide (NOS2 or iNOS) could be involved in the pathogenesis of SLE [4,5,6,7,8]. An inflammatory stimulus triggers iNOS production, further orchestrating inflammation, angiogenesis, and tissue fibrosis [9,10]. iNOS overexpression triggers cellular responses activating cell signaling pathways, oxidative damage, cell necrosis or apoptosis, and alteration of the immune response in patients with SLE [6,11]. Human iNOS activation in response to cytokines is regulated by inducible factor of hypoxia 1 (HIF-1) [9,10]. HIF regulates angiogenesis and the secretion of inflammatory cytokines in SLE by adapting to a hypoxic environment [12,13,14]. However, knowledge of the metabolic pathways that regulate the effects of iNOS activation on kidney function is limited.

SLE affects several organs and tissues, such as the skin, kidneys, joints, blood cells, heart, lungs, and brain [6,8,15]. Skin and mucous involvement includes skin rashes, photosensitivity, mucosal ulcers, and alopecia. Patients with SLE frequently develop lupus nephritis (LN), which can evolve to end-stage renal disease [5,7,8]. NL development is multifactorial, involving genetic susceptibility, environmental factors, and systemic inflammation [4,16,17,18]. Based on these findings, the present study explored the relationship between NOS2-associated inflammation profiles and SLE phenotypes.

## 2. Materials and Methods

### 2.1. Study Participants

The present study was a prospective, case control study developed over the last four years (2019–2023), with the patients selected from those who presented at the dermatology departments of the Clinical Hospital of Nephrology “Carol Davila” and Clinical Hospital “Victor Babes” in Bucharest, Romania. The study protocol was approved by the Ethics Committee of the Clinical Hospital of Nephrology “Carol Davila” (13/08.05.2019). The inclusion criteria were age older than 18 years old, with adequate nutritional status; healthy subjects were enrolled in the control group and patients with SLE in the other groups. The exclusion criteria were age younger than 18 years old, pregnancy, tobacco use, vitamin or antioxidants use, or stage 5 chronic kidney disease; any cardiovascular, hepatic, thyroid, gastrointestinal, or oncological disease; and any viral or bacterial infections in the previous three months. The study included a group of 86 subjects with cutaneous and hematological SLE, a group of 73 subjects with LN, and a group of 60 healthy subjects. SLE diagnosis was established according to Systemic Lupus International Collaborating Clinics/American College of Rheumatology criteria. The disease activity was based on the clinical Systemic Lupus Erythematosus Disease Activity Index (SLEDAI). LN subjects were diagnosed by biopsy puncture and histological exam according to KDIGO guidelines. In the SLE group, patients with stable treatment for at least six months (non-steroidal anti-inflammatory drugs, corticosteroids, hydroxychloroquine, immunosuppressant drugs such as azathioprine, mycophenolate mofetil, antihypertensive therapy) were included. 

The characteristics of the groups are presented in Table 1. They were similar in sex and age (*p* > 0.05). SLEDAI was statistically significant higher in the LN patients than in the SLE patients (*p* < 0.05). No statistical differences were detected in the clinical presentation of SLE and LN subjects (*p* > 0.05). SLE subjects presented statistically significantly higher serum creatinine, lower eGFR, and higher albuminuria compared with the LN group (*p* < 0.05) and control group (*p* < 0.05). When comparing the SLE and control groups, serum creatinine, eGFR and albuminuria did not vary statistically significantly (*p* > 0.05). 

### 2.2. Laboratory Determinations

The blood samples were collected from all the study participants after they signed an informed consent form at inclusion in the study. The collection was performed after 12 h of fasting, using a holder-vacutainer system. Centrifugation of the blood samples was performed at 3000× *g*, for 10 min after one hour maintaining the blood at room temperature. The sera were separated and frozen at −80 degrees before analyzing them. We excluded the hemolyzed, icteric, lactescent, or microbiologically contaminated samples.

HIF-1a and HIF-2a (ng/mL) were assessed using a sandwich ELISA Kit (MyBioSource, San Diego, CA, USA), a solid phase sandwich enzyme-linked immunosorbent assay. The antibodies were coated onto the microwells and incubated with the patient’s serum, and afterward, the HIF protein was captured by the coated antibody. Following extensive washing, an HIF detection antibody was added to detect the captured HIF protein. An HRP substrate, TMB, was added to develop the color. The magnitude of the optical density for this developed color was proportional to the quantity of HIF-1a with respect to the HIF-2a protein.

An inducible nitric oxide synthase (NOS2, iNOS E.C. 1.14.13.39) Activity Assay Kit (Colorimetric; CUSABIO Technology CSB-E08148h) was determined by ELISA sandwich, a serum quantitative determination of NOS2 human protein with analytical sensitivity (0.225 UI/mL, assay range 0.9 UI/mL–60 UI/mL), using TECAN analyzer.

VEGF (synonyms: VEGF-A, VPF) concentrations (pg/mL) were determined using the human Quantikine solid phase sandwich enzyme immunoassay ELISA kit DVE00 RD Systems, Minneapolis, USA; analytical sensitivity 9 pg/mL, assay range 15.6–2000 pg/mL. MMP-2 (E.C. 3.4.24.24, synonyms: 72-KDa type 4 collagenase, 72-KDa gelatinase, gelatinase A), and the concentration (ng/mL) was quantified using the human RayBiotech solid phase sandwich immunoassay ELISA kit ELH-MMP-2, analytical sensitivity 3.5 pg/mL, assay range 3.5-800 ng/mL. MMP-9 (E.C. 3.4.24.35, synonyms: 92-KDa gelatinase, 92-KDa type 4 collagenase, gelatinase B) concentrations (ng/mL) were determined using the human RayBiotech solid phase sandwich immunoassay ELISA kit (ELH-MMP-9, analytical sensitivity 10 pg/mL, assay range 10-6000 pg/mL). TSP-1 (synonym THBS) concentrations (ng/mL) were determined using the human Quantikine solid phase sandwich immunoassay ELISA kit (DTSP10, RD Systems, Minneapolis, MN, USA, with analytical sensitivity 0.944ng/mL and detection range 7.8–500ng/mL). Soluble VEGFR1/FLTI (E.C. 2.7.10, sVEGFR1, RTKs, soluble fms-like tyrosine kinase-1) concentrations (ng/mL) were quantitatively measured by solid-phase quantitative sandwich ELISA kit (CSB-P17498, which is commercially available; Cusabio, Houston, TX, USA, with analytical sensitivity of 0.039 ng/mL; assay range 0.156–10 ng/mL). All these laboratory parameters were colorimetrically detected at 450 nm using the TECAN analyzer.

### 2.3. Statistical Analysis

We used means and standard deviations for data presentation. Data between groups were compared using either ANOVA with Tukey’s post hoc test or the Kruskal–Wallis test with Dunn’s post hoc test for normally and non-normally distributed data. The relationship between the studied parameters was assessed by Pearson’s correlation coefficient, but before the assessment, data normality was evaluated by the Kolmogorov–Smirnov test. The level of significance (*p*) chosen was 0.05 (5%), and the confidence interval was 95% for hypothesis testing, with the corresponding ethical approval code.

## 3. Results

### 3.1. Inflammation-Associated Factors Studied in SLE and Control Groups

CRP, an important inflammation marker, presented serum levels less than 1.0 mg/dL in the control group. Its serum levels between 3 and 10mg/dL were considered low-grade inflammation and between 10 and 50 mg/dL as high-grade inflammation, given the levels found in the SLE and LN groups. NOS2, an enzyme involved in inflammatory processes, was overexpressed in the SLE and LN groups compared to controls. HIF-1a and HIF-2a, transcription factors with decisive roles in inflamed tissue adaptation to hypoxia, were overregulated in the studied SLE groups compared to the control group. High levels of VEGF in the SLE and LN groups versus controls could explain the angiogenic phenotype of autoimmune disease preceded by chronic inflammation. MMP-2 and MMP-9, metalloproteinases with essential effects on the reorganization of the extracellular matrix, showed significant increases in the SLE groups compared to controls. TSP-1, a cellular matrix protein involved in the development and resolution of inflammatory processes, presented significant lower levels in the LN group versus the SLE and control groups. Soluble VEGFR-1, a key factor in angiogenesis, was intensively consumed in the LN group compared to the SLE group. All data are presented in Table 2.

### 3.2. Inflammation-Associated Ffactors and Renal Function

The contribution of inflammation to organ impairment, specifically renal damage and LN development, was demonstrated by analysis of inflammation parameters in relation to eGFR (Table 3) and albuminuria (Table 4). We determined the aforementioned levels and parameters by eGFR (mL/min/1.73mp): A1 –>90, A2 – 60–90, A3 – 45–60, and A4 – 30–45 (Table 3); and by albuminuria (mg/L): A1 –<30, A2 – 30–300, and A3 – >300 (Table 4).

CRP, NOS2, HIF-1a, HIF-2a and VEGF serum levels increased significantly, while eGFR decreased. MMP-2 increased significantly, while eGFR was greater than 45 mL/min/1.73 mp and decreased significantly when eGFR was less than 45 mL/min/1.73 mp. TSP-1 and sVEGFR-1 serum levels decreased significantly with eGFR.

CRP, NOS2, HIF-1a, HIF-2a, and VEGF serum levels increased significantly with albuminuria. MMP-2 presented the highest level in albuminuria of 30–300mg/L. TSP-1 and sVEGFR-1 serum levels decreased significantly with albuminuria.

### 3.3. Interplay between Inflammation-Associated Factors and Renal Impairment in SLE

We evaluated the relationship between inflammation markers and renal damage in LN (Table 5). CRP and NOS2 correlated strongly positively with decreased eGFR and increased albuminuria. HIF-1a correlated positively only with albuminuria increases, while HIF-2a correlated with none of the markers. VEGF and metalloproteinases correlated positively with albuminuria and had a weak correlation with low significance with eGFR. TSP-1 and sVEGFR-1 had weak, positive correlations of low significance with eGFR and a weak, negative correlation with no statistical significance with albuminuria.

## 4. Discussion

SLE is a multisystemic disease, with hematological, skin, heart, lung, kidney, joint, and cerebral involvement and with a complex pathogenicity, intensively studied in previous decades [6,8,15]. The present study evaluated the inflammatory profile orchestrated by inducible nitric oxide synthase in systemic lupus erythematosus and lupus nephritis, showing that renal lesions could be a consequence of inflammation-associated hypoxia. Over the last century, CRP was considered a marker of the extent and severity of inflammation in SLE, but currently, increasing experimental data have suggested that in SLE the role of CRP requires a different approach. Our results showed that, in SLE and LN, serum CRP levels did not correlate with disease progression. The inability of CRP to reflect inflammatory activity in SLE could be explained by IFN1 gene overexpression or dysregulation, inhibitory effects of IFN1 IL-6/IL-1beta-induced CRP gene transcription, CRP’s proinflammatory/anti-inflammatory contradictory effects, the existence of several conformational isoforms, the presence of anti-CRP antibodies, polymorphisms (such as rs1205) in the CRP gene, multiple locations of protein synthesis (hepatocytes, kidney cells, neural cells, respiratory epithelial cells, adipocytes, leukocytes), and local conditions (acid microenvironment, NO release, IFN activation, proinflammatory cytokines, urea, heat, inflamed tissue, calcium, phospholipase A2, transcriptional activation of STAT3, C/EBP and NF-kB) [2,19,20,21,22,23,24,25,26]. Many published reports investigating CRP as a biological response modifier in IFN-dependent conditions have shown that monomeric anti-CRP antibodies are associated with increased frequency and stimulation of IL-6 and TNF alpha production in LN [27]. The irreversible processing of native pentamer CRP in monomer isoforms could serve as a buffer mechanism that locates destructive proinflammatory actions at the site of inflammation, thus protecting against organ involvement in SLE [23,24].

Monomeric CRP induces the secretion and release of NOS2, suggesting a potential mechanism of amplified catabolism in the injured tissue [6,27]. The results of the present study are in accordance with this statement; we found that NOS2 is overexpressed in LN and SLE compared to controls, and the association of NOS2 and eGFR with albuminuria was very strong. These data suggest that the synthesis and secretion of NOS2 in response to inflammatory stimuli present in SLE and LN could contribute to glomerular and vascular damage through the formation of NO and ONOO-, alteration of enzymatic activity, and amplification of neoepitopes in autoantigens [6,11,28]. A meta-analysis showed the involvement of NOS2 in SLE pathogenesis by higher expression of NOS2 at the level of mRNA in SLE compared to controls; significant differences in NOS2 at the protein level between SLE and controls; inconsistent variations in serum nitrites/nitrates between patients and controls; association between NOS2 levels and tissue damage; NO’s effect on Th1/Th2 balance in autoimmune diseases; the positive association between nitrogen species levels and SLE activity; the positive correlation between the increase in NOS2 activity and the severity of SLE; the increase in NO and NOS2 expression in keratinocytes, epithelial cells, and renal tissue cells in patients with SLE; the promotion of helper T cells by NOS2; and the effect of NOS2 inhibitors on SLE [6]. All these data suggest that NOS2 promotes SLE organ involvement and its progression.

Increased activity of NOS2 and activation of HIFs are important features of acute and chronic inflammation in autoimmune diseases. Hypoxia causes an increase in NOS2 expression, and HIF1 is crucial for processing the NOS2 gene [6]. In the current study, HIF-1a and HIF-2a, transcription factors with decisive roles in the adaptation of inflamed tissue systems to hypoxia, are overregulated in SLE and LN. HIF-1a is an essential contributor to the decline in kidney function in these patients. These results suggest that the oxidative imbalance induced by albumin suppresses prolyl hydroxylases to accumulate HIF-1α, (mediating profibrotic effects in renal tubular cells) and HIF-2a (maintaining glomerular barrier integrity) [29,30,31]. Hypoxia influences the inflammatory reactions through transcriptional response programs mediated by HIFs and thus promotes angiogenesis and vascular permeability [6,32].

The data obtained in the present study showed that, in hypoxic and inflammatory conditions, patients with SLE and LN present imbalances between soluble proangiogenic and antiangiogenic circulating factors compared to controls. Patients with LN have increased levels of VEGF, MMP-2, and MMP-9, low values for TSP-1 and sVEGFR, and significant associations of these molecular factors with eGFR and albuminuria. However, in these patients, the angiogenic stimulators are overregulated (VEGF, MMP-2, MMP-9), the angiogenic suppressors are sub-regulated (TSP-1, sVEFGR), and these variations are associated with a decline in kidney function. VEGF levels were increased in our study in patients with SLE and NL, probably through transcriptional overregulation and HIF posttranscriptional stabilization [6,30,33]. VEGF is involved in nephrin activity modulation and in the maintenance of the integrity of glomerular slit diaphragms, with important roles in glomerular and endothelial lesions and in different glomerulopathies and in modulation in impaired renal function [34,35]. VEGF, VEGFR-1, VEGFR-2, trombomoduline, and angiopoetin-2 induce endothelial lesions in active LN [36]. Reduced TSP-1 was detected in patients with severe clinic manifestations in SLE [37]. Recent studies evaluating the association between MMPs and renal function in LN showed a strong correlation of MMP2 expression in glomeruli and increased SLE activity scores with decreased renal function [38,39,40]. Changes in the expression and activity of MMPs have been described in a number of glomerulopathies. An increase in proteolytic activity in the glomerulus induced development of proteinuria in murine SLE [32,34,41]. MMPs regulate cell turnover, modulate various growth factors, and participate in the progression of tissue fibrosis and of tubular epithelial cell apoptosis in a variety of kidney diseases, causing local inflammation, fibrosis, and hemodynamic changes [42].

In summary, in SLE and NL, the magnitude of inflammatory processes influences the progressive decrease in the filtration capacity of the kidney. CRP bioactivities in IFN-dependent diseases could be explained by the unifying, recently developed concept of regulating the secretion and activation of the CRP system, which can ensure the localization of proinflammatory destructive effects at the site of inflammation [22,23,24]. In the inflammation microenvironment, NOS2-induced nephrotoxicity mediated by reactive N–species is associated with albuminuria in our study. NOS2’s capacity of regulating the activation and induction of HIFs and NO-mediated hypoxia are important elements in chronic inflammatory disorders and in autoimmune diseases [43]. Our data showed that overregulation of HIF-1a and HIF-2a in NL occurred progressively as albuminuria increased. The two major active isoforms, HIF-1α and HIF-2α, play non-redundant roles in adaptation to hypoxia [30]. HIFs accumulating in hypoxic tissue could stimulate angiogenic responses and the interaction between proangiogenic (VEGF, MMP2/9) and antiangiogenic factors (sVEGFR, TSP-1) and promote the deterioration of kidney function in patients with LN.

In SLE patients, it is very important to identify very early the patients at risk for LN development to avoid invasive interventions and to establish effective medical interventions. The present study is the first one in the literature that debates an actual topic in international research: the inflammatory profile in SLE, orchestrated by inducible nitric oxide synthase. SLE and LN are complicated diseases, characterized by complex interactions between variables that cannot be satisfyingly described by quantitative relations. We described at one point (SLE subjects with chronic, stable treatment) the relationship among biochemical serum parameters, eGFR, and inflammatory profiles using only simple regression. We consider this fact a limitation of the study, and a multivariable regression analysis that includes several potential markers should be presented in a further study. Although a wide range of molecular parameters were analyzed, a limitation of our study was that it followed patients with SLE and LN for a period of four years who were only undergoing chronic stable immunosuppressant treatment. More limitations could be considered, including the lack of analysis according to lupus nephritis type.

## 5. Conclusions

Inflammation, hypoxia, and disturbed angiogenesis are closely associated with the decline of kidney function in SLE and LN. Inflammatory renal phenotypes, increased hypoxia, stimulation of angiogenesis, and inactivation of the factors involved in the resolution of inflammation are the main pathogenesis elements that induce the development of LN and decrease the filtration rate in SLE subjects.

## Figures and Tables

**Table 1 jpm-13-00934-t001:** Groups characteristics.

Characteristics	SLE	LN	Control	*p* Significance
Men: women	17/59	11/62	10/50	>0.05
Age (years old)	43.0 ± 6.5	44.1 ± 5.6	41.7 ± 6.5	>0.05
SLEDAI	9.4 ± 4.1	12.9 ± 3.9	-	-
Disease duration (years)	5.7 ± 1.4	6.5 ± 1.1	-	-
Clinical presentation (%)				
Constitutional manifestations				
Fatigue	37.4	39.7	-	>0.05
Fever	11.2	9.5	-	>0.05
Weight loss	7.1	5.7	-	>0.05
Cutaneous-mucosal lesions				
Vespertilio	38.4	33.7	-	>0.05
Photosensitivity	33.9	31.4	-	>0.05
Hyperkeratosis/cutaneous atrophy	27.1	22.8	-	>0.05
Depigmentation	14.5	12.6	-	>0.05
Digital ulcerations	8.2	7.3	-	>0.05
Mucosal ulcerations	11.4	9.8	-	>0.05
Alopecia	24.1	21.9	-	>0.05
Musculoskeletal manifestations				
Arthritis/arthralgia	39.7	36.4	-	>0.05
Myositis	21.1	17.2	-	>0.05
Systolic pressure (mm Hg)	11.9 ± 1.6	13.1 ± 1.4	11.7 ± 2.3	>0.05
Diastolic pressure (mm Hg)	6.8 ± 0.8	7.1 ± 0.8	6.9 ± 0.9	>0.05
Laboratory data				
Leukocytes (cells/mmc)	3700 ± 2103	4022 ± 1051	5426 ± 1210	>0.05
Hemoglobin (g/dL)	10.7 ± 1.4	10.1 ± 1.1	13.1 ± 1.3	>0.05
Urea (mg/dL)	28.5 ± 11.7	36.0 ± 12.5	22.5 ± 7.6	>0.05
Creatinine (mg/dL)	0.9 ± 0.11	1.27 ± 0.19	0.68 ± 0.19	>0.05
eGFR (ml/min/1.73mp)	79.4 ± 16.8	44.7 ± 12.6	87.5 ± 14.2	>0.05
Hematuria (sw—RBC/camp)	5 ± 2	37 ± 8	1 ± 1	>0.05
Leukocyturia (sw—leuc/camp)	10 ± 2	9.3 ± 1.9	3.8 ± 2.9	>0.05
Albuminuria (mg/L)	0.012 ± 0.0003	1900 ± 1175	0.0001 ± 0.0001	<0.001
Albumin (g/dL)	3.72 ± 0.32	3.41 ± 0.29	4.11 ± 0.37	>0.05

Vespertilio—an erythematous rash in butterfly wings on the nose and dorsal face and at the zygomatic level.

**Table 2 jpm-13-00934-t002:** Inflammation-related factors in studied groups.

Parameters	SLE Group (A, *n* = 86)	LN Group (B, *n* = 73)	Control Group (C, *n* = 60)	*p*1	*p*2
CRP (mg/dL)	20.4 ± 11.3	24.6 ± 9.8	1.0 ± 1.0	0.009	A vs. B = 0.084 A vs. C = 0.003 B vs. C = 0.000
NOS2 (U/L)	43.7 ± 13.8	75.9 ± 25.3	12.6 ± 1.5	0.002	A vs. B = 0.001 A vs. C = 0.006 B vs. C = 0.000
HIF-1a (ng/mL)	108.9 ± 26.4	168.2 ± 56.3	56.8 ± 16.1	0.017	A vs. B: = 0.024 A vs. C = 0.024 B vs. C = 0.001
HIF-2a (ng/mL)	4.82 ± 1.7	7.0 ± 2.8	1.41 ± 0.70	0.008	A vs. B = 0.014 A vs. C = 0.001 B vs. C = 0.000
VEGF (pg/mL)	377.3 ± 88.3	541.5 ± 103.5	118.6 ± 20.6	0.009	A vs. B = 0.011 A vs. C = 0.014 B vs. C = 0.001
MMP-2 (ng/mL)	904.7 ± 142.6	1272.7 ± 406.4	453.3 ± 84.4	0.001	A vs. B = 0.019 A vs. C = 0.026 B vs. C = 0.002
MMP-9(ng/mL)	492.3 ± 101.4	808.2 ± 275.8	168.2 ± 26.9	0.029	A vs. B = 0.012 A vs. C = 0.002 B vs. C = 0.003
TSP-1(ng/mL)	716.2 ± 201.4	351.7 ± 105.3	1109.0 ± 102.9	0.027	A vs. B = 0.003 A vs. C = 0.033 B vs. C = 0.0001
sVEGFR-1 (ng/mL)	10.7 ± 4.3	6.3 ± 2.8	14.9 ± 2.2	0.026	A vs. B = 0.0046 A vs. C = 0.012 B vs. C = 0.002

LN-lupus nephritis; CRP-C-reactive protein; NOS2-inducible nitric oxide synthase; HIF-hypoxia inducible factor; VEGF-vascular endothelial growth factor; MMP-matrix metalloproteinase; TSP-thrombospondin; sVEGFR-soluble vascular endothelial growth factor receptor; *p*-significance level, p1-triple comparison of the groups, p-pairwise comparison of the groups

**Table 3 jpm-13-00934-t003:** Summary of inflammation-related parameter levels in LN patients according to eGFR.

eGFR (mL/min/1.73mp)					
Parameters	> 90(A1)	89–60 (A2)	59–45 (A3)	30–44 (A4)	*p* significance
Subjects number	18	23	20	12	
CRP (mg/L)	11.4 ± 6.9	17.1 ± 13.4	24.9 ± 17.2	34.5 ± 22.9	A1 vs. A2 = 0.019
					A1 vs. A3 = 0.001
					A1 vs. A4 = 0.0001
					A2 vs. A3 = 0.022
					A2 vs. A4 = 0.001
					A3 vs. A4 = 0.17
NOS2 (U/L)	31.4 ± 16.9	52.8 ± 15.1	89.1 ± 42.3	91.6 ± 53.7	A1 vs. A2 = 0.021
					A1 vs. A3 = 0.001
					A1 vs. A4 = 0.0001
					A2 vs. A3 = 0.017
					A2 vs. A4 = 0.021
					A3 vs. A4 = 0.054
HIF-1a (ng/mL)	66.4 ± 5.6	102.8 ± 27.3	201.4 ± 34.6	216.5 ± 44.2	A1 vs. A2 = 0.001
					A1 vs. A3 = 0.001
					A1 vs. A4 = 0.0001
					A2 vs. A3 = 0.012
					A2 vs. A4 = 0.018
					A3 vs. A4 = 0.044
HIF-2a (ng/mL)	2.9 ± 0.8	5.1 ± 3.9	6.2 ± 4.0	10.7 ± 4.1	A1 vs. A2 = 0.031
					A1 vs. A3 = 0.019
					A1 vs. A4 = 0.0001
					A2 vs. A3 = 0.046
					A2 vs. A4 = 0.032
					A3 vs. A4 = 0.045
VEGF (pg/mL)	178.3 ± 37.6	299.9 ± 107.2	602.8 ± 186.2	878.2 ± 272.7	A1 vs. A2 = 0.001
					A1 vs. A3 = 0.0001
					A1 vs. A4 = 0.0001
					A2 vs. A3 = 0.001
					A2 vs. A4 = 0.001
					A3 vs. A4 = 0.017
MMP-2 (ng/mL)	494.2 ± 85.9	902.5 ± 301.3	1408.3 ± 402.6	516.5 ± 174.9	A1 vs. A2 = 0.001
					A1 vs. A3 = 0.001
					A1 vs. A4 = 0.051
					A2 vs. A3 = 0.002
					A2 vs. A4 = 0.002
					A3 vs. A4 = 0.001
TSP-1(ng/mL)	1001.4 ± 140.5	422.2 ± 142.4	313.1. ± 123.0	248.2 ± 117.3	A1 vs. A2 = 0.001
					A1 vs. A3 = 0.0001
					A1 vs. A4 = 0.001
					A2 vs. A3 = 0.027
					A2 vs. A4 = 0.031
					A3 vs. A4 = 0.27
sVEGFR-1 (ng/mL)	13.9 ± 2.3	7.2 ± 2.1	6.1 ± 2.4	5.7 ± 1.4	A1 vs. A2 = 0.012
					A1 vs. A3 = 0.014
					A1 vs. A4 = 0.022
					A2 vs. A3 = 0.047
					A2 vs. A4 = 0.071
					A3 vs. A4 = 0.22

eGFR-estimated glomerular filtration rate; LN-lupus nephritis; CRP-C-reactive protein; NOS2-inducible nitric oxide synthase; HIF-hypoxia inducible factor; VEGF-vascular endothelial growth factor; MMP-matrix metalloproteinase; TSP-thrombospondin; sVEGFR-soluble vascular endothelial growth factor receptor; *p*-significance level.

**Table 4 jpm-13-00934-t004:** Summary of inflammation-related parameter levels in LN patients according to albuminuria.

Parameters	Albuminuria (mg/L)	<30 (A1)	30–300(A2)	>300 (A3)	*p* Significance
Subject number		19	40	14	
CRP (mg/L)		14.2 ± 7.1	23.9 ± 15.1	34.5 ± 17.1	A1 vs. A2 = 0.027 A1 vs. A3 = 0.001 A2 vs. A3 = 0.018
NOS2 (U/L)		31.2 ± 14.2	69.0 ± 42.0	104.8 ± 48.1	A1 vs. A2 = 0.011 A1 vs. A3 = 0.0001 A2 vs. A3 = 0.007
HIF-1a (ng/mL)		68.3 ± 10.1	147.4 ± 41.3	254.9 ± 51.2	A1 vs. A2 = 0.022 A1 vs. A3 = 0.0001 A2 vs. A3 = 0.032
HIF-2a (ng/mL)		3.1 ± 1.5	6.2 ± 3.3	11.0 ± 4.5	A1 vs. A2 = 0.012 A1 vs. A3 = 0.014 A2 vs. A3 = 0.047
VEGF (pg/mL)		221.2 ± 99.4	456.8 ± 193.4	897.1 ± 303.6	A1 vs. A2 = 0.019 A1 vs. A3 = 0.001 A2 vs. A3 = 0.002
MMP-2 (ng/mL)		614.5 ± 241.9	1112.3 ± 498.6	621.3 ± 611.6	A1 vs. A2 = 0.002 A1 vs. A3 = 0.0001 A2 vs. A3 = 0.001
TSP-1(ng/mL)		872.6 ± 166.3	367.6. ± 118.6	284.3 ± 119.1	A1 vs. A2 = 0.002 A1 vs. A3 = 0.0001 A2 vs. A3 = 0.57
sVEGFR-1 (ng/mL)		12.2 ± 3.2	6.6 ± 2.2	5.0 ± 1.6	A1 vs. A2 = 0.035 A1 vs. A3 = 0.001 A2 vs. A3 = 0.043

A2, A2, A3-albuminuria, LN-lupus nephritis; CRP-C-reactive protein; NOS2-inducible nitric oxide synthase; HIF-hypoxia inducible factor; VEGF-vascular endothelial growth factor; MMP-matrix metalloproteinase; TSP-thrombospondin; sVEGFR-soluble vascular endothelial growth factor receptor; *p*-significance level.

**Table 5 jpm-13-00934-t005:** Correlation analysis between inflammation–related molecules and renal impairment in LN.

Parameters	eGFR		Albuminuria	
	*r*	*p*	*r*	*p*
CRP	0.23	0.051	0.41	<0.01
NOS2	0.71	0.01	0.93	<0.01
HIF-1a	0.27	>0.05	0.91	<0.01
HIF-2a	0.17	0.05	0.36	0.05
VEGF	0.32	0.05	0.67	0.001
MMP-2	0.14	0.05	0.61	*p* < 0.01
MMP-9	0.22	0.05	0.72	*p* < 0.001
TSP-1	0.14	0.05	−0.19	*p* > 0.05
sVEGFR-1	0.26	0.05	−0.22	*p* > 0.05

LN-lupus nephritis; eGFR-estimated glomerular filtration rate; CRP-C-reactive protein; NOS2-inducible nitric oxide synthase; HIF-hypoxia inducible factor; VEGF-vascular endothelial growth factor; MMP-matrix metalloproteinase; TSP-thrombospondin; sVEGFR-soluble vascular endothelial growth factor receptor; *r*-correlation coefficient; *p*-significance level.

## Data Availability

All data are presented in the manuscript.

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
