# Peer review of "The Inflammatory Profile Orchestrated by Inducible Nitric Oxide Synthase in Systemic Lupus Erythematosus"

_jpm, 2023, doi:10.3390/jpm13060934_

Round 1

Reviewer 1 Report

The manuscript by Ene et al. is a case-control study aiming to compare enzymatic activity of NOS2 and serum levels of HIF-1a, HIF-2a, VEGF, MMP-2, MMP-9, TSP-1 and sVEGFR in patients with SLE, lupus nephritis and healthy controls. The manuscript is well and clearly written. I have a few questions/comments for the authors.

1. It is not clear at what time point of the disease the blood samples were taken for the analysis of the mentioned parameters.

2. Since the authors stated in the manuscript they recorded data about ongoing treatment for each patient, it would be interesting to see what effect the therapy has on the mentioned laboratory parameters (NOS2, HIF-1a, HIF-2a, VEGF, MMP-2, MMP-9, TSP-1 and sVEGFR).

3. In Table 1, it would be good to express SLEDAI as median +/- standard deviation, similar to age.

4. The term "Vespertilio" in Table 1 needs to be explained.

5. Looking at the data in Table 1, it could be concluded that the only parameter that differentiates patients with SLE, lupus nephritis and controls is albuminuria. However, it is quite clear that there are also differences in other variables, eg creatinine and eGFR (for example, the difference between patients with SLE and lupus nephritis). Perhaps it would make sense here to also show two p values, similar to Table 2 (p1; p2 - A vs. B, A vs. C, B vs. C).

6. It is necessary to state the limitations of the research.

7. Only two authors are listed on this manuscript, and it is a fairly large sample of patients and controls. Also, not even in the "Acknowledgments" did the authors single out anyone. Is it possible that this whole complex process was done by only two people?

Author Response

Dear Reviewer,

Thank you for giving me the opportunity to submit a revised draft of my manuscript titled

The inflammatory profile orchestrated by inducible nitric oxide synthase in systemic lupus erythematosus to Journal of Personalized Medicine. We appreciate the time and effort that you have dedicated to providing your valuable feedback on the manuscript. We are grateful for the insightful comments on the paper. We have been able to incorporate changes to reflect most of your suggestions provided. We have highlighted the changes within the manuscript.

Here is a point-by-point response to the reviewers’ comments and concerns.

  1. It is not clear at what time point of the disease the blood samples were taken for the analysis of the mentioned parameters.the mentioned parameters.

The blood samples were collected from all the study participants, after signing the informed consent at the inclusion in the study.

  1. Since the authors stated in the manuscript they recorded data about ongoing treatment for each patient, it would be interesting to see what effect the therapy has on the mentioned laboratory parameters (NOS2, HIF-1a, HIF-2a, VEGF, MMP-2, MMP-9, TSP-1 and sVEGFR).

In the SLE group, patients with stable treatment for at list six months (non-steroidal anti-inflammatory drugs, corticosteroids, hydroxychlorochine, immunosuppressant drugs such as azathioprine, mycophenolate mophetil, antihypertensive therapy).

  1. In Table 1, it would be good to express SLEDAI as median +/- standard deviation, similar to age.

We expressed SLEDAI as recommended.

  1. The term "Vespertilio" in Table 1 needs to be explained.

Vespertillio (erythematous rash in butterfly wings on nose dorsal face and at zygomatic level) was explained under Table 1.

  1. Looking at the data in Table 1, it could be concluded that the only parameter that differentiates patients with SLE, lupus nephritis and controls is albuminuria. However, it is quite clear that there are also differences in other variables, eg creatinine and eGFR (for example, the difference between patients with SLE and lupus nephritis). Perhaps it would make sense here to also show two p values, similar to Table 2 (p1; p2 - A vs. B, A vs. C, B vs. C).

We presented the groups characteristics in Material and Methods section:  The groups were similar as sex and age (p>0.05). SLEDAI was statistically significant higher in LN patients compared with SLE patients (p<0.05). No statistical differences were detected in clinical presentation of SLE and LN subjects (p>0.05). SLE subjects presented statistically significant higher serum creatinine, lower eGFR and higher albuminuria when compared with LN group (p<0.05), respectively control group (p<0.05). When com-pared SLE and control groups, serum creatinine, eGFR and albuminuria did not vary sta-tistically significant (p>0.05).

  1. It is necessary to state the limitations of the research.

We mentioned the limitations of the research in the last paragraph of Discussion.

  1. Only two authors are listed on this manuscript, and it is a fairly large sample of patients and controls. Also, not even in the "Acknowledgments" did the authors single out anyone. Is it possible that this whole complex process was done by only two people?

We have completed the Acknowledgments.

Reviewer 2 Report

Some issues are present that, when addressed, would further improve the clarity of the manuscript. The followings are the major comments:

1. Objective. I suggest removing statement that the study aims to explore the potential mechanisms by which iNOS can be involved in lupus nephritis development. This study can only explore association between markers and disease phenotype, not the actual mechanism behind the association.

2.Methods. Please specific the time frame during which the study was conducted.

3.Methods. Association between markers and clinical characteristics are complex. I therefore suggest performing a multivariable regression analysis that includes several potential markers. This would help confirm the validity of such associations, if they exist.

4.Methods. Please comment on sample size; whether it has been estimated and if it is sufficient for the present analysis.

no comment.

Author Response

Dear Reviewer,

Thank you for giving me the opportunity to submit a revised draft of my manuscript titled

The inflammatory profile orchestrated by inducible nitric oxide synthase in systemic lupus erythematosus to Journal of Personalized Medicine. We appreciate the time and effort that you have dedicated to providing your valuable feedback on the manuscript. We are grateful for the insightful comments on the paper. We have been able to incorporate changes to reflect most of your suggestions provided. We have highlighted the changes within the manuscript.

Here is a point-by-point response to the reviewers’ comments and concerns.

  1. I suggest removing statement that the study aims to explore the potential mechanisms by which iNOS can be involved in lupus nephritis development. This study can only explore association between markers and disease phenotype, not the actual mechanism behind the association.

We removed the mentioned statement.

  1. Please specific the time frame during which the study was conducted.

We mentioned the frame time in Material and Methods.

  1. .Methods. Association between markers and clinical characteristics are complex. I therefore suggest performing a multivariable regression analysis that includes several potential markers. This would help confirm the validity of such associations, if they exist.

SLE and LN are complicated diseases, characterized by complex interactions between variables that cannot be satisfyingly described by quantitative relations. We described at one point (SLE subjects with chronic, stable treatment) the relation between biochemical serum parameters, eGFR and inflammatory profile using only simple regression. We consider this as a limitation of the study, and a multivariable regression analysis that includes several potential markers would be presented in a future study.

  1. Please comment on sample size; whether it has been estimated and if it is sufficient for the present analysis.

SLE and LN are complicated diseases, characterized by complex interactions between variables that cannot be satisfyingly described by quantitative relations. We described at one point (SLE subjects with chronic, stable treatment) the relation between biochemical serum parameters, eGFR and inflammatory profile orchestrated by inducible nitric oxide synthase. We consider the sample size sufficient for the present analysis.

Round 2

Reviewer 1 Report

Please specify the duration of SLE at the time of sampling.

Author Response

Dear Reviewer,

Thank you for giving me the opportunity to submit a revised draft of my manuscript titled

The inflammatory profile orchestrated by inducible nitric oxide synthase in systemic lupus erythematosus to Journal of Personalized Medicine. We appreciate the time and effort that you have dedicated to providing your valuable feedback on the manuscript. We are grateful for the insightful comments on the paper. We have highlighted the changes within the manuscript.

  1. Please specify the duration of SLE at the time of sampling.

We added the disease duration in Table 1.

Reviewer 2 Report

no comment

no comment

Author Response

Dear Reviewer,

Thank you for giving me the opportunity to submit a revised draft of my manuscript titled

The inflammatory profile orchestrated by inducible nitric oxide synthase in systemic lupus erythematosus to Journal of Personalized Medicine. We appreciate the time and effort that you have dedicated to providing your valuable feedback on the manuscript.